# The Impact of Financial Development on Industrial Upgrading Based on the Analysis of Intermediation Effect and Threshold Effect

**Guihuan Yan** [1,2] **and Yi Chen** [3,*]

1. Ecology Institute of Shandong Academy of Science (China-Japan Friendly Biotechnology Research Center, Shandong Academy of Sciences), Jinan 250000, China; shengtaisuoygh@163.com
2. Shandong Technology Innovation Center of Carbon Neutrality, Jinan 250000, China
3. Faculty of Economics and Management, Qilu University of Technology (Shandong Academy of Sciences), Jinan 250000, China
* Correspondence: chenyiyi0906@163.com; Tel.: +86-198-6217-3009

**Abstract:** Accelerating industrial upgrading is essential for sustainable development. This paper aims to study how financial development affects industrial transformation and upgrading. First, the financial development index (fin) and the industrial upgrading index (htec, indu) are built using the entropy value approach, which is based on panel data from 30 provinces and cities in China from 2010 to 2020; second, using government intervention as the threshold variable, a fixed effects model and a threshold-effects model are utilized to empirically examine the non-linear link between financial development and industrial sophistication; and finally, the mechanism of financial development on industrial upgrading is examined using the mediating effect model, with science and technology innovation serving as the mediating variable. The study found that financial development has a positive contribution to high-tech industries (htecs). There is a U-shaped non-linear relationship between financial development and industrial advancement (indu). Finance has a stronger effect on promoting industrial upgrading via the intermediary role of science and technology innovation (tec). There is a significant double threshold effect between finance and industrial upgrading. Based on this, this paper puts forward countermeasure suggestions from the perspectives of financial development and scientific and technological innovation. It provides a basis for decision making to realize China's industrial upgrading and helps in sustainable development in the economy and society.

**Keywords:** financial development; industrial upgrading; science and technology innovation; intermediary effect; the threshold effect

## 1. Introduction

The 14th National People's Congress Government Work Report states the following: to promote the transformation and upgrading of traditional industries. The 20th Party Congress report also called for the promotion of a new type of industrialization and the acceleration of the construction of a strong manufacturing country. China is at a crucial juncture in its efforts to change its economic development model and achieve sustainable development. In this context, industrial upgrading is of great concern. Development in the financial industry has provided new opportunities for industrial upgrading, which is facilitated through financial support and investment guidance.

Goldsmith was one of the first scholars to introduce the concept of financial development; his theory of financial structure laid the foundation for the subsequent study of financial phenomena and the formation of other financial theories. The functional view of finance has been proposed by scholars such as Bodie and Levine; they redefined the concept of financial development, and a stable improvement in financial functions was included as an aspect of promoting financial development. The first domestic research on

the development in the financial system was conducted by Professor Chin-Shian Bai. He pointed out that the expanded financial function system, improved financing efficiency, and perfect financial services are a series of factors that can widely stimulate growth in financial development [1–4]. This paper is predicated on the accessibility of data and duly considers the evolution of financial institutions, financial instruments, and financial markets. The chosen metrics encompass the banking, security, and insurance sectors.

Upgrading refers to companies in the value chain that gain higher competitiveness by improving their technology and marketability. According to Gereffi, industrial upgrading describes the transition of labor-intensive industries to capital-, knowledge-, and technology-intensive ones. Kaplinsky argues that to determine whether an industry or product is upgrading, one should examine changes in both the relative price and market share of the product. If both rise and fall at the same time, the upgrade is achieved, but if both rise and fall, the upgrade cannot be confirmed by a simple judgment [5,6]. This paper measures the level of industrial upgrading in terms of industrial advances and the share of high-tech industries.

Since the concept of scientific and technological innovation was created, scholars at home and abroad have had different definitions of it. Schumpeter was a pioneer of the concept, defining innovation as the introduction of four aspects: new products, new production methods, new markets, and new raw materials. Freeman believes that science and technology innovation depends not only on individual firms or entrepreneurs but also on a national innovation system to achieve it. At the same time, under the arrangement of the national system, the institutions will continuously promote knowledge innovation, transfer, and application, which in turn will lead to nationwide science and technology innovation. Cooke creatively extends the concept of technological innovation to the regional level, pointing out that enterprises, research institutes, universities, and other institutions in the region interact to form an organizational system of innovation [7–9].

Through an empirical analysis of financial development and industrial upgrading, the main goal of this paper is to clarify the mechanism of financial influence on industrial upgrading. It also looks closely at the impact of financial development indicators on the indicators of industrial upgrading. Based on the findings of the study, we propose countermeasures in terms of financial subsectors and technological innovation, hoping to promote industrial upgrading and green and sustainable development.

The main contributions of this paper are as follows: (1) In the previous literature, most of the papers measured industrial upgrading in terms of industrial advancement and industrial rationalization, while this paper considers the share of high-technology industries. This paper argues that high-tech industries better reflect the development trend of industrial upgrading, which has lower resource consumption and environmental pollution and is conducive to promoting green and sustainable development and achieving sustainable development goals. (2) The impact of financial development on industrial upgrading is explored through the mediating variable of technological innovation, and the threshold variable of government intervention is added to verify the non-linear relationship between financial development and industrial upgrading, which provides strong evidence for the government facilitation of industrial upgrading.

Development that satisfies current demands without jeopardizing the requirements of present or future generations is referred to as sustainable development [10]. Through the accomplishment of 17 objectives for sustainable development in the economic, social, and environmental domains, the SDGs seek to promote cogent global economic, social, and environmental progress. Financial development and industrial upgrading are important means for achieving sustainable development. They play a key role in accelerating sustainable development. Financial development can reduce poverty, improve the quality of life, promote economic growth, and protect the environment. It provides financial support, risk management, and financial innovation for sustainable development. Industrial upgrading has the potential to facilitate economic transformation and innovation, foster sustainable industrial development, and enhance resource efficiency and environmental advantages.

Specifically, financial development can help achieve SDG 1 (no poverty), SDG 7 (affordable and clean energy), SDG 8 (economic growth), SDG 9 (industry, innovation, and infrastructure), and SDG 13 (climate action). Industry upgrading can help achieve the SDG 9, SDG 12 (responsible consumption and production), and SDG 13 goals. Therefore, the relationship between financial development, industrial upgrading, and sustainable development means that they are closely linked, and they can contribute to each other and to sustainable development.

This paper comprises the following components: Section 2 comprises a comprehensive review of the literature; Section 3 presents the theoretical analysis and research hypotheses; and Section 4 outlines the research design, encompassing model construction, variable design, and data sources. Section 5 is the empirical analysis. The last section presents the conclusion and policy recommendations.

## 2. Literature Review

### 2.1. Financial Development Concept and Measurement

Most scholars who have studied the issue of financial development have based their research on the perspective of economic growth. By the beginning of the last century, Schumpeter had already focused on the ability of financial development to promote economic growth and technological progress, an assertion that played an important role in the rise in post-war financial development theory [11]. Subsequently, the theory of financial development was developed with the release of Goldsmith's seminal book, Financial Structure and Financial Development [1]. By researching the amount of financial and economic growth in emerging nations, McKinnon and Shaw came up with their renowned "financial repression" and "financial deepening" ideas. Based on previous research, Robert proposed a functional theory of financial development, arguing that finance should be understood from a functional perspective. Brown further enriched the theory of financial development via the perspective of financial functions, arguing that finance can reduce financial risks and help alleviate frictions in financial markets via innovation [12–15].

Regarding the measurement indicators of the financial development level, foreign scholars have studied them previously. Goldsmith proposed the famous financial-related ratio indicator, which measures the degree of financial development in a country by the ratio of the value of all of its financial assets to the value of its physical assets [1]. Subsequently, Robert constructed indicators to measure the level of financial development in terms of depth, bank, and private [14]. McKinnon proposed using the volume of money as a share of total economic activity to measure the level of financial development [16]. Later, as the stock and insurance markets gained prominence in the financial markets, stock market capitalization and insurance depth were added to the indicators used to evaluate financial development.

### 2.2. The Connotation and Influencing Factors of Industrial Upgrading

In the 17th century, William Petty, a classical British political economist, pointed out that labor would shift between industries depending on the income of each sector, which is also considered to be the earliest theory of industrial structure. By analyzing output and labor force data for twenty countries, Clark devised the famous allotment-Clark theorem. In this theory, it is stated that the development in the economy and the increase in the per capita income level will lead to the transfer of labor among industries, that is, the rising proportion of secondary and tertiary industries' output value in the GNP represents the pattern of industrial transformations. The hypothesis of the dominating industry's diffusion effect was put forth by Rostow: in all stages of economic development, the expansion of the dominant sector will bring economic growth, thus generating a diffusion effect [17,18].

Swicki introduced technological progress, factor costs, and international trade into his model of industrial structural change, and he found that the most important factor affecting industrial change is technological progress. Wurgler points out that finance facilitates industrial restructuring from the perspective of capital allocation efficiency.

Through the study of capital allocation efficiency in different industries, it was found that its improvement is positively correlated with the degree of industrial development, thus optimizing the allocation of social means of production and promoting industrial upgrading [19,20]. Demand factors, supply factors, and international factors can all have an impact on industrial upgrading. In this paper, the proportion of high-tech industries and the proportion of three industries are used as indicators to measure industrial upgrading.

### 2.3. The Impact of Financial Development on Industrial Upgrading

In terms of theory, foreign scholars have previously studied the impact of financial development and innovation on industrial development, and early Chinese scholar Shijin Liu believed that the focus of financial development should be put on promoting industrial development, and that financial innovation that is conducive to industrial progress should be promoted [21]. A further study by Justin Yifu Lin found that financial development facilitates the restructuring of factor endowments [22]. The contribution of financial deepening and financial structure to industrial upgrading is dynamic [23]. For the measurement of financial development indicators, most scholars study the impact of digital inclusive finance, technology finance, and green finance on industrial structure upgrading [24–26]. Jiexi Zhu and Junjiang Li constructed composite indicators of the financial development level from the perspectives of financial structure, scale, and efficiency using the entropy method [27]. Zhongqiao Li measured the level of financial development deepening using the year-end loan balance and deposit balance of financial institutions [28]. On the contrary, Xinqian Du quantified the degree of financial development based on direct and indirect financial dimensions [29].

For an empirical study, Yuantian Li and Yingming Xu employed panel vector autoregressive (PVAR) models and impulse response analysis to examine the effects of alterations in financial structure on industrial upgrading [30]. Xiaolong Li and Guanghe Ran empirically tested the impact of digital finance on industrial upgrading with the help of a two-way fixed effect model, mediating effects, and a threshold model, and concluded that digital finance development has a significant double threshold effect on industrial structure upgrading [31]. Development in digital inclusive finance also provides a new perspective for industrial upgrading, and there is a non-linear relationship between the impacts of both. There will be spatial spillover effects on the region and surrounding areas [32–35].

### 2.4. Study on the Threshold Effect of Financial Development and Industrial Upgrading

The "threshold effect", first identified by Grossman and Helpman in their study of international technology spillovers from the foreign trade route, is that when a region's level of economic development has not yet reached a certain threshold, the pull of international trade on economic growth is very limited. When the region's economic development level crosses this threshold, international trade has a strong pull on economic growth. Most scholars have now confirmed the role of financial development in promoting industrial transformation and upgrading, but some scholars believe that excessive development in finance may lead to the flow of resources to the financial sector and ignore the needs of the real economy, resulting in a mismatch of industrial resources and restricting advanced industrialization [36–38]. Based on the panel data of prefecture-level cities, Wenjin Tang discovered a bottleneck in the advancement in digital inclusive finance, a threshold effect, and a non-linear relationship with industrial upgrading [39]. Based on the threshold model, Rongjuan Tan and Qiyuan Lu concluded that there is a threshold effect on the depth of use and digitalization of financial inclusion [40]. In summary, the impact of financial development on industrial transformation and upgrading is not just a linear relationship; there is a non-linear impact of first inhibiting and then promoting or first promoting and then inhibiting [41].

*2.5. Research Related to Science and Technology Innovation and Industrial Upgrading*

Technological innovation has changed the production methods and processes of traditional industries, promoted the transformation of traditional industries to high-end, intelligent, and green industries, and provided new growth points and momentum for industrial upgrading [42]. On the one hand, from the input perspective, technological innovation can improve the level of industrial productivity and increase the economic benefits of industry; on the other hand, from the output perspective, it can promote an improvement in product quality and performance, thus increasing the value of products, improving market competitiveness, enhancing the status of industry in the international market, and promoting industrial upgrading. Aizhen Li et al. used a dynamic GMM and threshold model to empirically test the relationship between the three, and found that science and technology innovation played a mediating role and there was a non-linear relationship between it and industrial upgrading [43]. Other scholars have found that financial development can optimize the capital supply structure, reduce pollution, rationalize the distribution of production factors, improve production efficiency, and thus promote industrial upgrading via technological innovation [44–46].

For the measurement of science and technology innovation, the quantity of patent applications and the allocation of patent grants are widely acknowledged metrics in academia, and some scholars also use the ratio of technology market turnover to GDP, the ratio of weighted patent applications to total regional population, and the proportion of R&D to total manufacturing output to measure it [47–49]. Most scholars use advanced and rationalized industries to build the index system for measuring the upgrading of the industrial structure, while some scholars build the indexes from industrial efficiency, value-added products, and industrial structure [50–52].

In summary, most scholars have constructed comprehensive evaluation indexes for financial development, calculated the weights of each index using the entropy method, and studied the impact on industrial upgrading using mediating effects and spatial econometric models. The construction of industrial upgrading indicators is also only at the level of advances and rationalization, without considering the important role of high-tech industries. Based on data availability, this paper constructs financial development indicators from three representative industries, namely, banking, security, and insurance. Since the upgrading of industry is not just the increase in the ratio of the three industries, but the high-tech industry is the core force to promote industrial transformation, this paper replaces the traditional industrial rationalization index with the ratio of the high-tech industry and adds industrial advances to measure the level of industrial upgrading.

## 3. Theoretical Analysis and Hypothesis

*3.1. The Impact of Financial Development on Industrial Upgrading*

With the development and improvement in economics-related theories, the research perspective on financial development theory has become more diversified and the research results have become more abundant. The main developments are the financial structure theory, the deepening theory, the constraint theory, and the financial function theory. In the functional theory of finance, Levine identifies five basic functions of finance in the economy: resource allocation function, risk control function, promoting capital circulation function, optimizing enterprise management function, and facilitating the exchange and flow of commodities and services [3]. The performance of the five functions can promote technological innovation and capital accumulation, which in turn can promote industrial upgrading. In the theory of industrial structure, Kuznets refers to the primary, secondary, and tertiary industries as agriculture, industry, and services, respectively. The proportion of the agricultural sector has been diminishing, while at the same time, the share of the industrial sector has been increasing. Additionally, the proportion of the workforce engaged in the service sector has experienced a more substantial surge [53].

According to the theory of financial function, financial development has an impact on industrial upgrading mainly through capital formation and capital-directed mechanisms.

Banks and other financial institutions gather idle funds through savings and other means to form a certain scale of capital, which facilitates investment and financing for enterprises and provides more financial support for their expansion. Stocks, bonds, and other financial instruments have short financing times and high efficiency, alleviating the information asymmetry in the investment and financing processes and promoting industrial upgrading. However, at the same time, excessive development in the financial sector will lead to an insufficient supply of resources and funds for the real economy, which will lead to asset bubbles and credit risks, thus limiting the speed and scale of industrial transformation. Accordingly, the following hypotheses are formulated:

**Hypothesis 1.** *There is a threshold effect between financial development and industrial upgrading.*

**Hypothesis 2.** *Financial development promotes industrial upgrading.*

*3.2. The Effect of Innovation in Science and Technology on Industrial Upgrading*

Schumpeter improved and developed the innovation theory and clearly divided "innovation" into five types: developing new products, adopting advanced production methods, opening new markets, acquiring new production materials, and adopting new organizational methods [7].

Science and technology innovation can influence industrial upgrading by optimizing factor allocation and adjusting the demand structure, guiding the transfer of factors from inefficient industries to high-efficiency, high-value-added industries. It can promote development in new products, change the original market structure, guide consumers to adjust their own needs, gradually reduce their reliance on old products, and increase the purchase of new products. At the same time, through technological innovation, research into more energy-efficient and environmentally friendly production methods and products to meet consumer demand for environmental protection and energy conservation drive development in environmental industries and promote sustainable development. Accordingly, Hypothesis 3 is formulated:

**Hypothesis 3.** *Financial development promotes industrial upgrading via technological innovation.*

**4. Data and Methods**

*4.1. Model Construction*

In order to verify Hypothesis 1 and Hypothesis 2 and to explore the impact of financial development on industrial upgrading, the following model was constructed in this paper:

$$htec_{it} = \alpha_0 + \alpha_1 fin_{it} + \alpha_2 X_{it} + \lambda_t + \varepsilon_{it} \tag{1}$$

$$indu_{it} = \beta_0 + \beta_1 fin_{it} + \beta_2 X_{it} + \lambda_t + \varepsilon_{it} \tag{2}$$

To explore the threshold effect of both, Model (3) was constructed by adding the threshold variable of government intervention (gov), drawing on Hansen's study, to test the threshold effect of financial development on industrial sophistication [54]. gov is the threshold variable, $\gamma$ is the threshold value, $X_{it}$ is the control variable, and the degree of influence of financial development on industrial advancement is $\delta_1$ and $\delta_2$, respectively.

$$indu_{it} = \delta_0 + \delta_1 fin_{it}(gov_{it} \leq \gamma) + \delta_2 fin_{it}(gov_{it} > \gamma) + \delta_3 X_{it} + \lambda_t + \varepsilon_{it} \tag{3}$$

To test Hypothesis 3, the following mediating effect model was constructed in this paper: subscript $i$ denotes the $i$th province, subscript $t$ denotes year $t$, and $X_{it}$ is the set of control variables. *htec* is the explanatory variable of high-technology industry share, *indu* is the explanatory variable of industry advances, *fin* is the explanatory variable of financial development, and *tec* is the mediating variable of science and technology innovation. $\mu_1$

denotes the direct effect of financial development on high-tech industries, and $\gamma_1 \times \mu_2$ is the indirect effect.

$$tec_{it} = \gamma_0 + \gamma_1 fin_{it} + \gamma_2 X_{it} + \lambda_t + \varepsilon_{it} \tag{4}$$

$$htec_{it} = \mu_0 + \mu_1 fin_{it} + \mu_2 tec_{it} + \mu_3 X_{it} + \lambda_t + \varepsilon_{it} \tag{5}$$

Table 1 shows the meanings of the other variables:

**Table 1.** Variable names and meanings.

| Variable Symbols | Meanings |
| --- | --- |
| $\alpha_0, \beta_0, \delta_0, \gamma_0, \mu_0$ | Constant term |
| $\alpha_1, \beta_1, \delta_1, \delta_2, \gamma_1, \mu_1, \mu_2$ | Coefficient of the effect of explanatory variables on the explained variables |
| $\eta, \delta_3$ | Coefficient of influence of control variables on explanatory variables |
| $\lambda_t$ | Fixed effects |
| $\varepsilon_{it}$ | Residual term |

### 4.2. Variable Design

#### 4.2.1. Explained Variables

Most scholars have constructed the index system of industrial structure upgrading with industrial advances and rationalization, and this article measures the proportion of high-technology industry (htec) and industrial advances (indu) [55], because the upgrading of industry is greater than the increase in the proportion of three industries, and high-technology industries can more obviously accelerate the development trend of industrial transformation.

#### 4.2.2. Explanatory Variables

Drawing on the research of Xizhang Liu and Zeheng Yang, we constructed the financial development index system at the level of banking, security, and insurance industries, and used the entropy value method to calculate the weights of each index, as shown in Table 2 below [56].

**Table 2.** Financial development index system.

| Level 1 Indicator | Parameter Symbol | Weight |
| --- | --- | --- |
| Market value of shares | $X_1$ | 0.4690 |
| Premium income | $X_2$ | 0.1693 |
| Loan balance of financial institutions | $X_3$ | 0.1695 |
| Balance of deposits in financial institutions | $X_4$ | 0.1922 |

#### 4.2.3. Control Variables

Based on the existing literature studies [57], this article selects the level of economic development (eco), openness to the outside world (open), infrastructure (inf), and information technology (info) as the control variables. Variable descriptions are shown in Table 3.

### 4.3. Data Source

Due to the serious lack of industrial data in Tibet, this paper selects the data related to financial development and industrial upgrading from 30 provinces and cities in China from 2011 to 2020. The data were obtained from the China Statistical Yearbook, China Industrial Statistical Yearbook, Wind Database, China High-Technology Industry Statistical Yearbook, and Provincial Statistical Yearbooks, etc. We used Stata.17 software for the model and data processing.

**Table 3.** Variable descriptions.

| Variable Type | Variable Name | Variable Symbol | Variable Description |
|---|---|---|---|
| Explained variables | High-tech industry share | htec | High-tech operating income/industrial operating income |
| | Advanced industrialization | indu | Output value of the third industry/output value of the second industry |
| Explanatory variables | Financial development | fin | Entropy method |
| Intermediate variables | Technology innovation | tec | R&D investment intensity |
| Threshold variables | Government intervention | gov | Local budget expenditures/regional GDP |
| Control variables | Economic development level | eco | Gross regional product/regional population |
| | Degree of openness to the outside world | open | Total exports and imports/GDP |
| | Infrastructure | inf | Highway mileage/administrative area |
| | Informatization level | info | Telecommunications business volume/regional GDP |

## 5. Empirical Analyses

### 5.1. Descriptive Statistics

Table 4 displays the findings of the descriptive statistics for the key variables in this study, from which it can be seen that the mean level of financial development (fin) is 0.137, with a minimum value of 0.00183 and a maximum value of 0.96, indicating a large gap in the level of financial development. The mean value of high-technology industry share (htec) is 0.114 with a small standard deviation, which indicates that China's high-technology industry is developing faster and financial development has a certain promotion effect on industrial upgrading. The large standard deviation in the variables of industrial advances (indu) and infrastructure level (inf) may be due to the large changes in the value added of the tertiary sector and the level of infrastructure in the country from 2011 to 2020. Among the control variables, the level of information technology (info) has the smallest standard deviation of 0.0538, indicating that it has the smallest inter-provincial variation, which may be related to the intelligence, digitalization, and high-end quality of information in China in recent years.

**Table 4.** Descriptive statistics of main variables.

| Variables | Mean | Sd | Min | Max |
|---|---|---|---|---|
| htec | 0.114 | 0.0780 | 0.004 | 0.387 |
| indu | 1.325 | 0.730 | 0.527 | 5.297 |
| fin | 0.137 | 0.156 | 0.00183 | 0.960 |
| tec | 0.00257 | 0.00230 | 0.00003 | 0.0118 |
| eco | 1.578 | 0.438 | 0.464 | 2.799 |
| open | 0.274 | 0.290 | 0.00757 | 1.464 |
| inf | 0.943 | 0.502 | 0.0890 | 2.205 |
| info | 0.0581 | 0.0538 | 0.0147 | 0.285 |

Before conducting regression analysis, an LLC test was performed on the panel data to exclude pseudo-correlation cases, and the results showed that the *p*-value of the unit root test for all of the variables was less than 0.05, rejecting the hypothesis that all variables were non-stationary; so, the data were stationary for further regression analysis.

Table 5 reports the results of the variance inflation factor test for the main explanatory variables, with a mean VIF value of 2.46, which is less than 10, so there was no serious problem of multicollinearity between the variables.

**Table 5.** VIF test results.

| Variables | Vif | 1/Vif |
|---|---|---|
| fin | 3.480 | 0.288 |
| tec | 3.370 | 0.297 |
| eco | 3.080 | 0.324 |
| open | 2.480 | 0.402 |
| inf | 1.730 | 0.577 |
| indu | 1.720 | 0.581 |
| info | 1.360 | 0.737 |
| Mean | VIF | 2.460 |

*5.2. An Empirical Test of Financial Development Affecting Industrial Upgrading*

The results in Table 6 were obtained by regressing the two-way fixed effects model of financial development and industrial upgrading. Financial development was significantly and positively correlated with the share of high-tech industries at the 1% level, indicating that finance plays a certain role in promoting development in high-tech industries, and Hypothesis 2 was verified.

**Table 6.** Linear regression results of financial development and the share of high-tech industries and industrial advancement.

| Variables | htec | indu |
|---|---|---|
| fin | 0.113 *** | −0.733 * |
| | (0.030) | (0.379) |
| eco | 0.057 | −0.660 ** |
| | (0.044) | (0.309) |
| open | 0.069 * | −1.110 *** |
| | (0.040) | (0.255) |
| inf | 0.098 ** | 0.139 |
| | (0.037) | (0.278) |
| info | 0.071 | −0.878 |
| | (0.077) | (0.826) |
| | (0.029) | (0.237) |
| _cons | −0.099 * | 2.152 *** |
| | (0.052) | (0.381) |
| time/individual fixed | Yes | Yes |
| N | 300 | 300 |
| adj. $R^2$ | 0.66 | 0.83 |

Notes: *, **, and *** denote significance at the 10%, 5%, and 1% confidence levels, respectively; unmarked regression results are not significant, as follows.

Because China is still in a phase of transitional economic growth, development in the financial industry has not yet reached a high level, which may be the cause of the negative relationship between development in the financial industry and the economy and the advanced industry. The level of industrial progress is low, and the relevant auxiliary resources are insufficient. The level of external openness also has a dampening effect on industrial sophistication, probably because a higher level of external openness will intensify market competition and make enterprises focus more on cost reduction, and imperfect policies such as intellectual property protection will also lead to a lack of incentives for enterprises to achieve sophistication.

The above test found that financial development has a negative effect on industrial sophistication. In this paper, we further constructed Model (6) based on Model (2) by adding the quadratic term of financial development to test the U-shaped non-linear relationship between them.

$$indu_{it} = \theta_0 + \theta_1 fin_{it} + \theta_2 fin_{it}^2 + \theta_3 X_{it} + \lambda_t + \varepsilon_{it} \qquad (6)$$

From the results in Table 7, we can see that financial development has a significant negative correlation of 5% with industrial advances. The quadratic term of financial development is significantly positively correlated with industrial advancement at 1%, with the coefficient of the primary term being less than 0 and the coefficient of the secondary term being greater than 0. This indicates that the two are not simply linear and negative but show a U-shaped non-linear relationship, and there is an optimal level of influence between the two. When financial development is at a lower level, the advanced level of industry is low due to insufficient policy support, lack of capital, and a low technology level; when financial development is at a higher level, the effect of advanced industry is brought into play.

**Table 7.** Non-linear regression results of financial development and industrial advancement.

| Variables | indu |
|:---:|:---:|
| fin | −1.949 ** |
| | (0.826) |
| $fin^2$ | 4.427 *** |
| | (1.007) |
| eco | 0.448 *** |
| | (0.135) |
| open | 0.735 *** |
| | (0.171) |
| inf | −0.132 |
| | (0.0904) |
| info | 2.164 *** |
| | (0.697) |
| constant | 0.492 *** |
| | (0.154) |
| observations | 300 |
| R-squared | 0.41 |

Notes: **, and *** denote significance at the 5%, and 1% confidence levels, respectively; unmarked regression results are not significant, as follows.

*5.3. A Test of the Threshold Effect of Financial Development Affecting Industry Advances*

As shown in Table 8 and Figure 1, the first threshold is 0.172 and the second threshold is 0.358. The effect of financial development on industrial sophistication is not significant when the level of government intervention is less than 0.172. When the level of government intervention is greater than 0.172, the level of financial development effectively promotes industrial sophistication, indicating that each unit increase in the level of financial development increases industrial sophistication by 1.487 to 7.275, and the level of government support promotes industrial sophistication to some extent. The relationship between financial development and industrial advancement tends to suppress and then increase with increasing government support, and this discovery substantiates the non-linear association established in the preceding section. Notably, amidst the control variables examined, both the level of economic development and the extent of openness to external influences exhibit a noteworthy and affirmative impact on industrial progress, while the level of infrastructure and information technology do not have a significant effect on industrial advancement. As can be seen from Table 9, the self-sampling 300 tests in Stata were conducted to first verify the significance of the three thresholds, and the results showed that the three threshold tests were not significant, and the single and double thresholds were significant at the 1% level. Hypothesis 1 was verified.

**Table 8.** Threshold test of financial development for industry advancement.

| Variables | indu |
|---|---|
| first threshold | 0.172 |
| second threshold | 0.358 |
| gov ≤ 0.172 | −0.187 |
| | (0.322) |
| 0.172 < gov ≤ 0.358 | 1.487 ** |
| | (0.630) |
| gov > 0.358 | 7.275 *** |
| | (1.516) |
| eco | 0.361 *** |
| | (0.100) |
| open | −0.980 *** |
| | (0.273) |
| inf | 0.192 |
| | (0.323) |
| info | 0.408 |
| | (0.288) |
| constant | 0.673 *** |
| | (0.214) |
| observations | 300 |
| R-squared | 0.83 |

Notes: ** and *** denote significance at the 5% and 1% confidence levels, respectively; unmarked regression results are not significant, as follows.

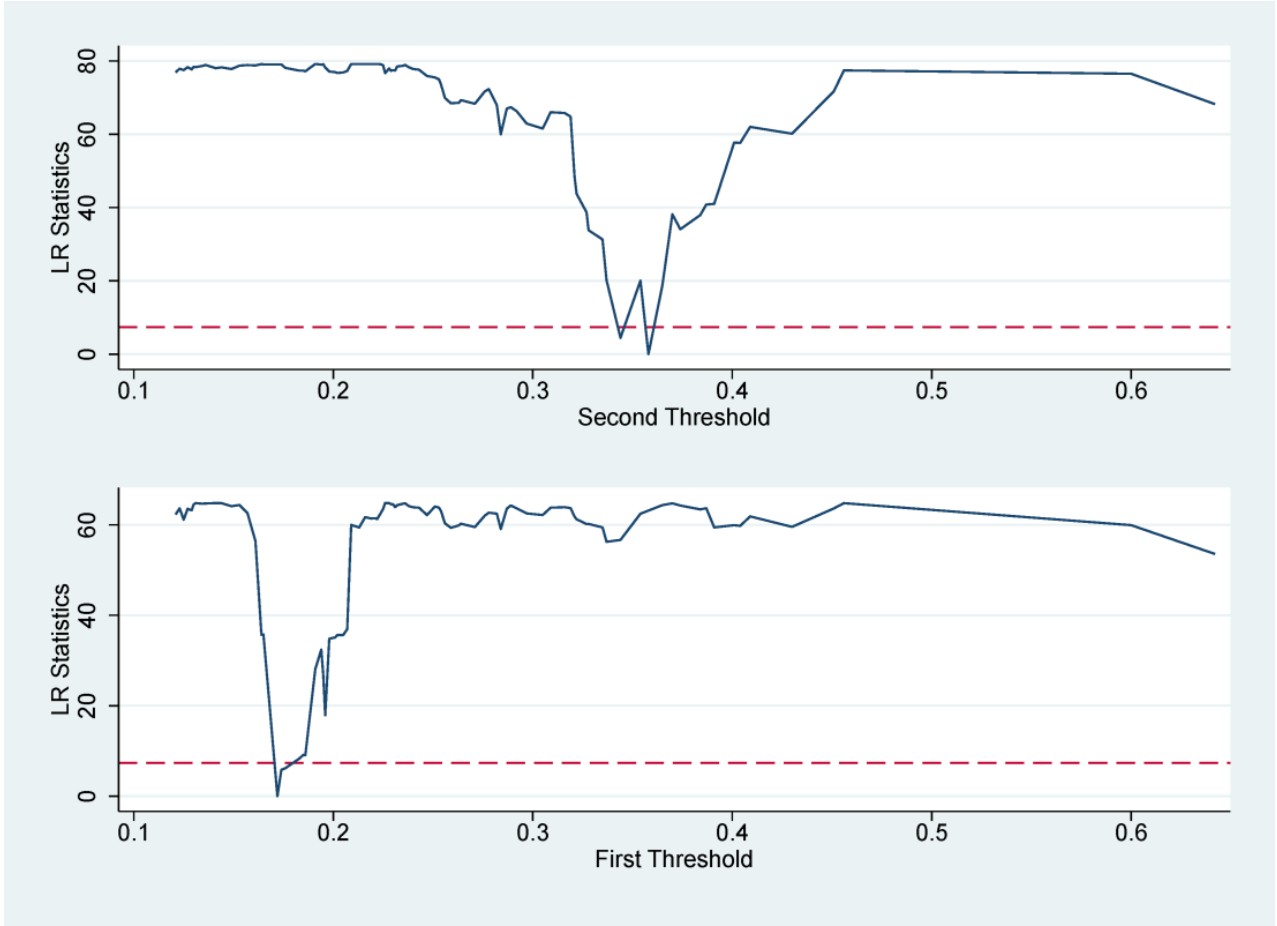

**Figure 1.** Results of two-threshold estimation.

**Table 9.** Results of the threshold effect test.

| Models | *p*-Value | 10% Threshold | 5% Threshold | 1% Threshold |
|---|---|---|---|---|
| Single threshold | 0.0033 *** | 31.83 | 39.4474 | 46.4793 |
| Dual threshold | 0.0033 *** | 30.7 | 35.3403 | 49.7571 |

Notes: *** denote significance at the 1% confidence levels, respectively; unmarked regression results are not significant, as follows.

### 5.4. A Test of the Mediation Effect of Financial Development on the Share of High-Tech Industries

This paper has explored the mechanism of the effect of financial development on high-tech industries from the perspective of science and technology innovation, and the empirical results were obtained by constructing a mediating effect model for regression, as shown in Table 10. The first column is a two-way regression of the fixed effects of financial development on high-tech industries, which leads to the conclusion that developments in finance (fin), economy (eco), openness to the outside world (open), and the level of infrastructure (inf) effectively contribute to a development in high-tech industries. The regression analysis presented in the second column indicates a statistically significant positive association between financial development and innovation in the domain of science and technology. The third column shows that the impact of financial development on high-tech industries is significantly positively correlated at the 1% level under the mediating role of technology innovation, and the impact of both decreases from the original 0.113 to 0.081, indicating that technology innovation plays a partly mediating role.

**Table 10.** Test results of the mediating effect of technology innovation between financial development and the share of high-tech industries.

| Variables | (1) htec | (2) tec | (3) htec |
|---|---|---|---|
| fin | 0.113 *** | 0.006 *** | 0.081 *** |
|  | (0.020) | (0.001) | (0.021) |
| eco | 0.057 ** | 0.000 | 0.055 ** |
|  | (0.025) | (0.001) | (0.024) |
| open | 0.069 *** | −0.000 | 0.070 *** |
|  | (0.026) | (0.001) | (0.026) |
| inf | 0.098 *** | 0.002*** | 0.087 *** |
|  | (0.019) | (0.001) | (0.018) |
| info | 0.071 | −0.002 | 0.082 |
|  | (0.055) | (0.002) | (0.058) |
| tec |  |  | 5.150 *** |
|  |  |  | (1.647) |
| _cons | −0.231 *** | −0.003 * | −0.216 *** |
|  | (0.059) | (0.002) | (0.058) |
| N | 300 | 300 | 300 |
| adj. $R^2$ | 0.97 | 0.96 | 0.97 |

Notes: *, **, and *** denote significance at the 10%, 5%, and 1% confidence levels, respectively; unmarked regression results are not significant, as follows.

From the table, the size of this intermediary effect is 0.0309 (obtained by multiplying the coefficient of the second column fin with the coefficient of the third column tec), which is about 27.34% of the total effect of financial development on high-tech industries (obtained by dividing the above intermediary effect by the coefficient of the first column fin). The findings of this study suggest that financial development may facilitate growth in high-tech industries via the mechanism of science and technology innovation. Furthermore, this mechanism can account for 27.34% of the overall impact of financial development on high-tech industries, with the indirect effect accounting for 38.15% of the direct effect (obtained via the coefficient of the third column fin of the intermediary effect ratio above), verifying the previous hypothesis.

Furthermore, using the Sobel test as shown in Table 11, the Sobel and Goodman test *p*-value < 0.01, indicating that the mediating effect was also confirmed and financial development will increase the share of high-technology industries via science and technology innovation.

**Table 11.** Sobel test.

|  | Coef | Z-Value | *p*-Value |
| --- | --- | --- | --- |
| Sobel | 0.0317 | 2.67 | 0.007 |
| Goodman test 1 | 0.0317 | 2.653 | 0.007 |
| Goodman test 2 | 0.0317 | 2.688 | 0.007 |
| Indirect effect | 0.0317 | 2.67038 | 0.005 |
| Direct effect | 0.0808 | 3.2802 | 0.001 |
| Total effect | 0.1125 | 5.06618 | $4.10 \times 10^{-7}$ |

*5.5. Robustness Tests*

Due to the excessive impact of the new crown outbreak on the economy starting in 2019, to avoid its influence on the empirical results and to ensure the robustness of the conclusions, this paper reduced the sample period to 2011–2018 and re-examined it. As shown in Tables 12 and 13, and the conclusions remained largely unchanged.

**Table 12.** Results of the test of the mediating effect between scientific and technological innovation in financial development and the share of high-technology industries.

|  | (1) htec | (2) tec | (3) htec |
| --- | --- | --- | --- |
| fin | 0.104 *** | 0.004 *** | 0.083 *** |
|  | (0.033) | (0.001) | (0.031) |
| eco | 0.030 | −0.001 | 0.033 |
|  | (0.028) | (0.001) | (0.027) |
| open | 0.089 *** | −0.001 | 0.095 *** |
|  | (0.021) | (0.001) | (0.022) |
| inf | 0.084 ** | 0.002 *** | 0.073 ** |
|  | (0.033) | (0.001) | (0.031) |
| info | 0.137 | −0.002 | 0.150 |
|  | (0.102) | (0.005) | (0.108) |
| tec |  |  | 5.355 *** |
|  |  |  | (1.882) |
| _cons | −0.167 ** | 0.001 | −0.171 ** |
|  | (0.082) | (0.002) | (0.080) |
| N | 240 | 240 | 240 |
| adj. R² | 0.97 | 0.96 | 0.97 |

Notes: **, and *** denote significance at the 5%, and 1% confidence levels, respectively; unmarked regression results are not significant, as follows.

**Table 13.** Sobel test for robustness.

|  | Coef | Z-Value | *p*-Value |
| --- | --- | --- | --- |
| Sobel | 0.0211 | 2.1490 | 0.0316 |
| Goodman test 1 | 0.0211 | 2.0990 | 0.0359 |
| Goodman test 2 | 0.0211 | 2.2040 | 0.0275 |
| Indirect effect | 0.0211 | 2.1493 | 0.0316 |
| Direct effect | 0.0827 | 2.6236 | 0.0087 |
| Total effect | 0.1037 | 3.3503 | 0.0008 |

**6. Conclusions and Policy Implications**

*6.1. Conclusions*

Drawing on scholarly research, this research article investigated the influence of financial development on every aspect of industrial upgrading using a two-way fixed effects model, mediating effects, and threshold effects models. The study employed panel data from 30 provinces and cities in China, spanning from 2011 to 2020. The empirical findings are as follows: (1) The development in finance has effectively promoted a development in high-tech industries, but the levels of finance, economy, and openness to the outside world significantly inhibit advanced industrialization. (2) Financial development has a U-shaped non-linear relationship that first inhibits and then increases industrial sophistication. In a period of economic transformation, finance is not enough to support the trend of advanced industrialization in China, and the financial industry helps to upgrade the industry when development reaches a certain level. (3) There is a double threshold between financial development and an advanced industrial structure. Under the influence of government intervention, the effect is initially insignificant, but as financial development crosses the threshold, the promotion effect on advanced industrialization becomes stronger and increases in significance. (4) With the inclusion of technological innovation, the direct influence of the financial sector on high-tech industries decreases, and technological innovation plays a partially intermediary role between the two. Economic development and the level of openness to the outside world change the influence on industrial advancement from negative to positive after adding mediating variables, and an improvement in infrastructure also significantly promotes a development in industrial advancement. (5) After shortening the sample period, financial development continues to promote industrial advancement under the influence of technological innovation as a mediator.

*6.2. Policy Implications*

(1) We should promote coordinated development in various subsectors of finance, and help with development in high-tech industries and industrial advancement at the level of credit, the capital market, and the insurance market. Commercial banks should support financial products that meet the local reality, build a financial service system that combines standardized products with regional specialties, and increase credit investment in the high-tech manufacturing industry. Capital market financial institutions can provide venture capital and equity investment support for innovative enterprises through the establishment of industrial funds and equity investment funds to promote industrial upgrading and technological innovation.

(2) The link between financial development and industrial upgrading is non-linear and U-shaped, suggesting that the influence of finance on industrial upgrading possesses a "double-edged sword" characteristic. As such, it is imperative to mitigate the idle financial capital bubble and minimize resource misallocation costs. The government and financial institutions should strengthen collaboration to provide quality financial services, and coordinate multiple industrial policies, such as talent policy and fiscal policy, according to the needs of industrial upgrading. Industrial policy can have a positive impact on industrial upgrading through a variety of means, such as resource allocation and guidance, technological innovation and R&D support, market access and exit, and coordinated regional development and industrial clusters, as well as industrial norms and standards development. Therefore, the government needs to formulate appropriate industrial policies to guide industrial upgrading according to the national development strategy and industrial development stage.

(3) To strengthen technological innovation and give full play to this intermediary role, the government can support enterprises to broaden their technological fields, conduct research and develop new technologies, and benchmark using international advanced levels to achieve industrial upgrading; promote the integration of finance and technology and their application in different fields, such as smart manufacturing, internet of things, etc.; encourage cooperation between financial institutions and technology companies to

jointly promote development in financial technology; provide more efficient and convenient services for industry; and enhance industrial efficiency and competitiveness. Under the financial structure dominated by indirect financing, commercial banks should increase capital investment in high-tech fields, include support for technology innovation enterprises in the assessment index, and also provide credit guarantees for enterprise financing to lower the cost of enterprise financing to improve the success rate of financing.

*6.3. Research Shortcomings and Future Directions*

In this paper, when selecting financial development indicators, due to the availability of data, only the banking, security, and insurance industries were considered, which has limitations in terms of indicator representativeness. Future research should evaluate various aspects of financial scale, structure, and efficiency. The scope of this paper is relatively narrow, focusing only on the impact of financial development on industrial upgrading in each province and city. Future research can expand the scope of research and study the relationship between the two at the national, prefecture, and city levels.

**Author Contributions:** Conceptualization, G.Y. and Y.C.; methodology, Y.C.; data curation, Y.C.; software, Y.C.; formal analysis, Y.C.; resources, G.Y.; project administration, G.Y.; supervision, G.Y.; writing—original draft, Y.C.; writing—review and editing, G.Y. All authors have read and agreed to the published version of the manuscript.

**Funding:** Jinan City 2022 Talent Development Special (grant no. 202228126) and Shandong Province Humanities and Social Sciences Project (grant no. 2022YYGL-49).

**Institutional Review Board Statement:** Not applicable.

**Informed Consent Statement:** Not applicable.

**Data Availability Statement:** The data involved in this study are all from public data.

**Conflicts of Interest:** The authors declare no conflict of interest.

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
