# Peer review of "The Impact of Financial Development on Industrial Upgrading Based on the Analysis of Intermediation Effect and Threshold Effect"

_sustainability, doi:10.3390/su15108364_

Round 1

Reviewer 1 Report

The manuscript entitled "The Impact of Financial Development on Industrial Upgrading Based on the Analysis of Intermediation Effect and Thresh old Effect" aims at analyzing the mechanism of financial development on the dimensions of industrial upgrading using the mediating effect model, and further, empirically examining the nonlinear relationship between financial development and industrial advancement using government intervention as the threshold variable.

To achieve the aim of the paper, the authors have used a panel of 30 provinces and cities in China from 2011 to 2020.

Before the paper can be published, the following major amendments are requiredQ

1. The paper's introduction is too short. In this section the paper's aim and objectives must be explained. Furthermore, the paper's core concepts must be defined so as to further support it. Last, a paragraph supporting the paper's rationale must be included as well.

2. The literature review of the paper must be reconsidered so as it includes the analysis of all the research objectives.

3. Concering the paper's results, a critical analysis must be added referring to similar papers of other cases so as the research results can be sharpened.

4. Last, the authors must add 1-2 paragraphs to support the paper's relation with sustainability, since there is not an obvious connection (except for a short reference in the paper's methodology).

Reviewer 2 Report

The manuscript " The Impact of Financial Development on Industrial Upgrading-Based on the Analysis of Intermediation Effect and Threshold Effect " has been submitted for publication in Sustainability Journal. The manuscript covers a well-known relationship between the financial and Industry sector in a dynamic way. However, the manuscript cannot present some drawbacks summarized below. The author(s) are strongly encouraged to carefully remove these drawbacks by following the suggestions for improvement. Particularly attention to results discussion and industry policy implication.

Reviewer 3 Report

Dear Authors,

Thank you for your paper. I found it very interesting and well-prepared. However, it requires some minor changes to be published in Sustainability.

My suggestions read as follows:

1. Please add a subsection, “Study contribution”, and briefly describe the contribution of your results to the literature.

2. Please add a scheme of methods to provide your approach to the broader audience.

3. Please add a table with nomenclature and symbols used in your equations. 

Best regards,

Reviewer 4 Report

The manuscript analyzes the mechanism of financial development on the dimension of industrial upgrading. Although the research question of this manuscript is worth investigating, there are still several issues that avoid me suggesting it for publication. Therefore, I would suggest major revision.

1: In the manuscript I saw many long sentences. Please revise them and make your sentence shorter and more understandable.

2: Please talk about the innovations of this study in abstract (in brief) and in introduction (last paragraph).

3: How your paper can influence on sustainability and sustainable development. You use the following literatures as well to explain how industrial development can influence sustainability.

doi:10.4028/www.scientific.net/AMM.253-255.244

doi:10.3390/su11051353

4: Table 3: standard deviation should be ±. Also in the table caption explain all abbreviations.

5: I did not see any nonlinear regression in your manuscript, however, you mentioned nonlinear results! Please make it clear

6: How to define the threshold?

7: please report R2 with two-digit accuracy.

8: I suggest to change some of your tables to graph. Also in the tables show the most important results in bold.

9: conclusion should be in separate part and include the most important numerical results.

Round 2

Reviewer 1 Report

The paper is much improved compared with the primary submission while all the issues raised by the reviewer are addressed. 

Reviewer 4 Report

Thank you for your effort. However, there are still several errors specially in your reference format. For example Line 115, L128 you mentioned a study with out any reference. This should be checked through whole manuscript. Your paper needs extensive English proof.  Therefore, I would suggest major revision again. 

Round 3

Reviewer 4 Report

Good luck for your future research.